# β-Carotene from the Alga *Dunaliella bardawil* Decreases Gene Expression of Adipose Tissue Macrophage Recruitment Markers and Plasma Lipid Concentrations in Mice Fed a High-Fat Diet

**DOI:** 10.3390/md20070433

**Published:** 2022-06-29

**Authors:** Nir Melnikov, Yehuda Kamari, Michal Kandel-Kfir, Iris Barshack, Ami Ben-Amotz, Dror Harats, Aviv Shaish, Ayelet Harari

**Affiliations:** 1The Bert W. Strassburger Metabolic Center, Sheba Medical Center, Ramat Gan 5262000, Israel; nir.melnikov@sheba.gov.il (N.M.); yehuda.kamari@sheba.gov.il (Y.K.); michal.kandel.kfir@sheba.gov.il (M.K.-K.); iris.barshack@sheba.gov.il (I.B.); dror.harats@sheba.gov.il (D.H.); aviv.shaish@sheba.gov.il (A.S.); 2The Sackler Faculty of Medicine, Tel-Aviv University, Tel-Aviv 6997801, Israel; 3N.B.T., Nature Beta Technologies Ltd., Eilat 8851100, Israel; amiba@bezequin.net; 4The Department of Life Sciences, Achva Academic College, Arugot 7980400, Israel

**Keywords:** vitamin A, β-carotene, obesity, adipose tissue, mice

## Abstract

Vitamin A and provitamin A carotenoids are involved in the regulation of adipose tissue metabolism and inflammation. We examined the effect of dietary supplementation using all-trans and 9-cis β-carotene-rich *Dunaliella bardawil* alga as the sole source of vitamin A on obesity-associated comorbidities and adipose tissue dysfunction in a diet-induced obesity mouse model. Three-week-old male mice (C57BL/6) were randomly allocated into two groups and fed a high-fat, vitamin A-deficient diet supplemented with either vitamin A (HFD) or β-carotene (BC) (HFD-BC). Vitamin A levels in the liver, WATs, and BAT of the HFD-BC group were 1.5–2.4-fold higher than of the HFD group. BC concentrations were 5–6-fold greater in BAT compared to WAT in the HFD-BC group. The eWAT mRNA levels of the *Mcp-1* and *Cd68* were 1.6- and 2.1-fold lower, respectively, and the plasma cholesterol and triglyceride concentrations were 30% and 28% lower in the HFD-BC group compared with the HFD group. Dietary BC can be the exclusive vitamin A source in mice fed a high-fat diet, as shown by the vitamin A concentration in the plasma and tissues. Feeding BC rather than vitamin A reduces adipose tissue macrophage recruitment markers and plasma lipid concentrations.

## 1. Introduction

Obesity is characterized by the excessive accumulation of fat in adipose tissue, which may impair health. Obesity is a major risk factor for cardiovascular diseases, type 2 diabetes, nonalcoholic fatty liver disease, and several types of cancers [1]. Although obesity has been recognized as a worldwide epidemic, the current interventions and treatments have many limitations. Several human studies have found correlations between obesity and low levels of serum or subcutaneous fat carotenoids [2,3,4,5,6,7]. 

Carotenoids are fat-soluble pigments synthesized in plants, fungi, bacteria, and algae. Provitamin A carotenoids, such as β-carotene (BC) and α-carotene, are cleaved in the body to produce vitamin A [8]. Vitamin A is an essential micronutrient for growth and development, vision, reproduction, and immunity in mammals. The main dietary sources of preformed vitamin A include meat and dairy products, while BC from plants serves as the primary provitamin A carotenoid in the human diet [8]. Vitamin A and provitamin A carotenoids are stored in the liver and adipose tissue. In these tissues, vitamin A and provitamin A carotenoids can be converted into the biologically active derivatives of vitamin A: retinol, retinal, and retinoic acid (RA) [9,10]. Although the distribution of vitamin A in different rodent adipose depots has been examined [11,12], to the best of our knowledge, the accumulation of provitamin A carotenoids in rodent WAT and BAT has not been investigated. 

In previous studies, we used the alga *Dunaliella bardawil* as a source for natural BC. This unicellular alga accumulates high BC concentrations (~10% of the dry weight), and it consists of two primary BC isomers: ~50% ATBC and ~50% 9-cis BC [13]. We have shown that dietary supplementation with 9-cis BC or a mixture of all-trans and 9-cis BC from *Dunaliella bardawil* can protect against atherogenesis and reduce plasma cholesterol concentrations in a mouse model of atherogenesis, while synthetic all-trans BC had the opposite effect [14]. In vitro and in vivo experiments have demonstrated that ATBC is enzymatically converted to all-trans RA (ATRA), while 9-cis BC is converted to both ATRA and 9-cis RA [15,16]. Both RA isomers are involved in the regulation of gene expression through the activation of nuclear receptors. Our studies have indicated that dietary *Dunaliella bardawil* supplementation may inhibit the development of obesity-associated pathological disorders, such as dyslipidemia, adipose tissue inflammation, diabetes, atherosclerosis, and fatty liver in *Ldlr^−/−^*, *Apoe^−/−^*, and *db/db* mice models [14,17,18,19]. These mouse models develop fatty liver when fed a high-fat diet [20,21,22]. Interestingly, another group recently reported that dietary BC, as an exclusive source of vitamin A, reduced plasma cholesterol concentration and atherosclerotic lesion size only when the production of vitamin A by enzymatic cleavage of BC is available [23,24]. This supports our hypothesis that the positive impact of *Dunaliella bardawil* supplementation in mice might be the result of the conversion of algal BC isomers into vitamin A derivates. 

So far, the effect of dietary BC supplementation on adiposity and adipose tissue metabolism has been investigated by relatively few studies utilizing non-obese animal models fed a standard diet containing vitamin A [25,26,27]. Moreover, the isomer composition of the BC used in these experiments has not been unspecified. Thus, it is likely that ATBC was the dominant isomer. In the current study, we sought to investigate the impact of dietary supplementation with ATBC and 9-cis BC-rich *Dunaliella bardawil* as the sole source of vitamin A on risk factors associated with obesity in a diet-induced obesity. 

## 2. Results

### 2.1. Body Weight, Energy Intake, and Tissue Mass

Body weight was measured throughout the 23-week experiment. The rate of body weight gain was higher in the HFD-BC group than in the HFD control group between weeks 7 and 9 (Figure 1A). However, no differences between the groups were observed in either NMR body composition analysis after 14 weeks (Appendix A) or body weight gain from the 10th week and onwards. At week 23, the WAT mass was 10% greater in the HFD-BC group than in the HFD group (*p* < 0.05). In contrast, the liver mass was similar (Figure 1C). Although we found a 5% increase in food intake in the HFD-BC group compared with the HFD group, this result could not be statistically analyzed because intake was measured per cage.

### 2.2. Tissue and Plasma Vitamin A Levels

Vitamin A levels (all-trans retinol) in the HFD-BC group were significantly higher in the liver, eWAT, iWAT, and iBAT, but not in the plasma, when compared with the HFD group (Table 1). As expected, vitamin A concentration in the liver, which is the primary vitamin A reservoir, was substantially higher than all adipose depots by two orders of magnitude (*p* < 0.0001) in both the HFD and HFD-BC groups. Within each group, vitamin A concentrations did not differ between eWAT, iWAT, and iBAT (*p* > 0.99).

### 2.3. Tissue and Plasma BC Concentrations

As expected, no isomers of BC were detected in the tissues of the mice in the HFD group supplemented exclusively with preformed vitamin A. On the other hand, both ATBC and 9-cis BC were detected in all analyzed samples of the HFD-BC group supplemented with BC given as *Dunaliella bardawil* powder (Table 2 and Figure 2). In the HFD-BC group, the plasma BC concentration was 0.48 ± 0.08 μg/mL, and the ATBC/9-cis BC ratio was 8.90 ± 1.45 (*n* = 5). Expectedly, the accumulation of total BC (μg/g tissue) in the liver (7.04 ± 1.45) of the HFD-BC group was 20-fold, 17-fold, and 3-fold greater than eWAT, iWAT, and iBAT, respectively (*p* < 0.001). ATBC/9-cis BC ratio in the liver (1.82 ± 0.19) was lower by 3.7-fold, 2.7-fold, and 5.8-fold than in eWAT, iWAT, and iBAT, respectively, here suggesting that the liver stores more 9-cis BC than the adipose tissue (*p* < 0.05). The ATBC:9-cis BC isomer ratio in *Dunaliella bardawil* powder was about 1:1. Nevertheless, ATBC was the dominant isomer present in the plasma and tissues. Liver BC/retinol ratio (0.04 ± 0) was lower by 23-fold, 22-fold, and 130-fold than in eWAT, iWAT, and iBAT, respectively (*p* < 0.05). Surprisingly, iBAT BC concentration was elevated by 5–6-fold compared to eWAT and iWAT. As a result, the ratio between BC to retinol was higher in iBAT than in eWAT and iWAT (Table 2). Furthermore, we found that the ATBC/9-cis BC ratio was elevated by 1.6- and 2.2-fold in iBAT compared with eWAT and iWAT, respectively (Table 2).

### 2.4. mRNA Levels of Inflammatory Cytokines and Transcriptional Regulators of Thermogenesis and Macrophage-Staining in Adipose Tissue

We examined the mRNA expression levels of inflammatory markers in eWAT and iBAT. The eWAT mRNA levels of *Mcp-1* and *Cd68*, which are markers for monocyte recruitment and tissue macrophages, were lower by 1.6- and 2.1-fold, respectively, in the HFD-BC group compared with the HFD group (Figure 3A). The mRNA levels of *Tnfα* and *Il-6* in eWAT and iBAT, as well as *Il-1β* in eWAT, were not different between the diet groups (Figure 3A,B).

We also assessed eWAT and iBAT mRNA levels of *Ucp1*, which is expressed in thermogenically active adipocytes, and of the transcriptional regulators of *Ucp1*: *Pparγ* and *Pgc1α* [28]. No differences were found between the mRNA levels of *Pgc1α* in iBAT or levels of *Ucp1* and *Pparγ* in both eWAT and iBAT between the HFD-BC and HFD groups (Figure 3B).

In immunohistochemical staining of adipose tissue, we detected macrophage infiltration in adipose tissue of high-fat fed mice, while in *Dunaliella*-treated mice no macrophages were identified. (Figure 3C,D).

### 2.5. Plasma Lipids, Leptin, and Adiponectin Concentration

The plasma cholesterol and triglyceride levels in the HFD-BC group were decreased by 28% and 30%, respectively, compared with the HFD group after 23 weeks of treatment (Figure 4A,B). On the other hand, no differences were observed in the plasma leptin and adiponectin concentrations between the HFD-BC and HFD groups (Figure 4C,D).

### 2.6. Glucose Metabolism, White Adipocyte Size, and NAFLD

No significant differences were identified between the HFD-BC and HFD groups for all the following parameters: glucose tolerance (IPGTT), fasting blood glucose levels, fasting plasma insulin concentrations, the white adipocytes area in eWAT, and NAFLD activity score (Appendix A).

## 3. Discussion

In the current study, we investigated the effect of BC-rich *Dunaliella bardawil* powder as the sole source of vitamin A on the development of obesity and its complications in mice. We have demonstrated that dietary BC maintains tissue and plasma vitamin A reservoir and that BC accumulation in BAT was elevated compared with WAT. Unexpectedly, we found that BC supplementation increased eWAT mass but reduced eWAT mRNA levels of macrophage recruitment markers and lowered plasma cholesterol and triglyceride levels in high-fat diet fed mice. Contrary to our hypothesis, BC treatment did not affect the mRNA levels of the genes involved in regulating adipose tissue thermogenic activity in iBAT or eWAT.

Our previous study [29] showed that BC supplementation decreased atherogenesis in an *Apoe*^−/−^ mouse model compared with a vitamin A deficient group. In that study, both groups were fed a chow diet. Since ApoE can affect lipid metabolism, it is crucial to study BC in a wild-type model. Therefore, in the current study, we investigated the effect of BC in C57BL/6 mice fed a high-fat diet compared with the vitamin A–fed group. The present study demonstrates that dietary BC maintains vitamin A levels (all-trans retinol) in the liver, adipose tissue, and plasma. Although vitamin A levels in the tissues of the HFD-BC group were elevated compared with the controls, it is important to note that we did not adjust the relative quantity of vitamin A (retinol equivalent) in the HFD BC group to the HFD control group. To the best of our knowledge, the estimated efficiency of BC isomer conversion to vitamin A in mice has not been evaluated [30]. Thus, adjusting the relative vitamin A content in the diets was not feasible. These results provide further evidence supporting BC’s ability to maintain tissue and plasma vitamin A levels as an exclusive dietary source in mice.

In addition to vitamin A, we examined the accumulation of BC in mouse liver and adipose tissue. Here, we show that higher levels of BC accumulated in mouse BAT compared with WAT. To the best of our knowledge, the accumulation of BC in mouse BAT has not been studied, and we are the first to report this finding. Since BC storage in mouse tissues occurs only upon exposure to remarkably high dietary concentrations, BC is usually undetected in the tissues of mice fed standard diets that contain low levels of BC [31]. Thus, mice lacking the BC degradation enzyme β-Carotene oxygenase 1 (*Bco1*) are typically utilized to examine BC accumulation in tissues. However, it is impossible to study the role of BC as a provitamin A carotenoid in this model. By fortifying the high-fat diet to wild-type mice, we were able to study whether the presence of BC in adipose tissue together with vitamin A confers beneficial effects over the presence of vitamin A alone. With a high dose of BC, we detected ATBC and 9-cis BC in several tissues, studying the effect of BC as a sole source of vitamin A on obesity development in mice. As expected, BC accumulation in the liver, which is the primary reservoir of vitamin A and BC in mammals, was higher than in adipose tissue. Nevertheless, the ratio between BC and vitamin A concentrations in adipose tissue was greater compared with the liver ratio, suggesting that BC metabolism is distinct in each tissue. Taken together, our results imply that BC metabolism in mice differs between the liver and adipose tissue and between BAT and WAT.

Next, we investigated the effect of BC enrichment on body weight and adiposity. Unexpectedly, we discovered that eWAT mass was higher in the HFD BC group compared with the HFD control group. However, we found no differences in adipocyte hypertrophy or body weight gain during the second half of the experiment. Conversely, a previous study showed that BC supplementation decreased the masses of the three WAT depots examined: inguinal, gonadal, and retroperitoneal [27]. Nevertheless, the study was performed on female mice (C57BL/6) fed a standard diet. Further studies may be required to determine whether the effect of dietary BC supplementation on WAT mass in mice is influenced by diet composition and gender.

Furthermore, we examined the inflammatory state in eWAT by analyzing the mRNA levels of inflammatory-related genes. The mRNA levels of *Mcp-1* and *Cd68* in eWAT were significantly lower (*p* < 0.05) in the HFD-BC group compared with the HFD control group. Moreover, macrophages were detected in adipose tissue of HFD-treated mice, while no macrophages were found in *Dunaliella*-treated mice. Similarly, in a previous study, we demonstrated that BC dietary supplementation reduced the mRNA level of *Mcp-1* in the mesenteric WAT of obese *db/db* mice [17]. The *Mcp-1* levels in adipose tissue and plasma of obese rodents are increased and promote adipose tissue macrophage infiltration [32,33,34]. Additionally, a study in humans demonstrated that the mRNA levels of *Mcp-1* and *Cd68* (macrophage marker) were elevated in subcutaneous WAT of obese individuals [35]. Notably, BC treatment decreased the expression of *Mcp-1* and the activity of *NFkB*, which regulates the expression of proinflammatory cytokines, in cultured adipocytes exposed to oxidative stress [36]. In addition to BC, its downstream cleavage product ATRA has been shown to have an anti-inflammatory effect [37]. Altogether, these results suggest that a BC-rich diet may reduce adipose tissue macrophage recruitment, possibly because of the presence of vitamin A, BC, or both.

Moreover, we sought to assess measures of obesity-associated hyperlipidemia. The plasma concentrations of cholesterol and triglycerides were lower in the HFD-BC group compared with the HFD control group. In previous studies, we have demonstrated that BC (given as *Dunaliella bardawil*) decreased plasma cholesterol concentrations in atherosclerosis mouse models (*Apoe^−/−^* and *Ldlr^−/−^*), as well as triglyceride levels in obese diabetic (*db/db*) mice [14,17,18,19]. Of note, oxidized (BC-free) *Dunaliella bardawil* powder did not reduce the plasma cholesterol concentration of *Ldlr^−/−^* mice [14]. A recent study has shown that BC, when given as the sole source of vitamin A, decreased plasma cholesterol levels in wild-type mice but not in *Bco1^−/−^* mice [23]. A follow-up study found that atherosclerotic lesion size and plasma cholesterol level were reduced by BC supplementation in *Ldlr^−/−^* mice but not in *Ldlr^−/−^/Bco1^−/−^* mice [24]. Overall, these observations suggest that the effect of BC on plasma lipids is probably mediated by *Bco1*-dependent cleavage of BC to form RA, a derivative of vitamin A that binds to and activates the nuclear receptors involved in regulating metabolic pathways [23,24].

Finally, we evaluated the thermogenic activity in eWAT and BAT by gene expression analysis. BC, when given as the only dietary source of vitamin A, did not affect the mRNA levels of genes involved in regulating thermogenesis (*Ucp1*, *Pparγ*, and *Pgc1α*) in the eWAT or BAT of the HFD-BC group compared with the HFD group. To the best of our knowledge, the effect of BC supplementation on the expression of *Ucp1*, which plays a major role in brown and beige adipocyte thermogenic activity, has only been examined in vitro [38,39] and in standard-diet-fed ferrets [26]. Remarkably, a review of several studies has concluded that high-fat diets may increase the expression of *Ucp1* in rodent BAT, likely reflecting an adaptive mechanism to excess caloric intake [40]. Hence, it is conceivable that high-fat diet feeding might have masked a feasible effect of BC on adipose tissue thermogenic capacity.

The positive effects of BC can be attributed to its role as a precursor of retinol, RA, and other retinoids. However, additional research is required to elucidate whether BC itself could act directly in this manner. It was shown by Lobo et al. that retinoid signaling and the expression of *Pparγ* in the WAT of vitamin A-deficient mice were both affected by BC treatment, while all-trans-retinol had no effect [41]. In addition, the study showed that in mature adipocytes, BC, not all-trans-retinol was metabolized to RA. Furthermore, we recently showed that serum and adipose tissue carotenoids, including BC, but not retinol, negatively correlated to many anthropometric and metabolic traits in humans [2].

The current study implies that BC metabolism in mouse WAT and BAT is different. Still, possible molecular pathways that may explain this difference have not been investigated. Moreover, we show that supplementing high-fat diet-fed obese mice with ATBC and 9-cis BC, as an exclusive vitamin A source, decreases eWAT macrophage infiltration markers and reduces cholesterol and triglyceride plasma levels.

## 4. Materials and Methods

### 4.1. Mice

Three-week-old male mice (C57BL/6JOlaHsd) were housed in plastic cages on wood-shaving bedding, under controlled ambient temperatures (23 ± 2 °C) and on a 12 h light/12 h dark cycle. The animals had free access to feed and water. The mice were killed after 23 weeks using isoflurane following a 15 h fast. Venous blood, liver, epididymal WAT (eWAT), inguinal WAT (iWAT), and interscapular BAT (iBAT) were collected. Blood samples were immediately centrifuged (10 min at 4 °C and 955 g), and the plasma was stored at −80 °C. Tissue samples were immediately snap-frozen in liquid nitrogen and stored at 80 °C until use. All procedures were performed in accordance with the Chaim Sheba Medical Center’s Guidelines for Animal Studies and approved by the Institutional Animal Ethics Committee (ethical approval code 1204/19).

### 4.2. Diet

We used a vitamin A-deficient, high-fat diet (20% of the calories from protein, 20% from carbohydrate, 60% from fat, D06040702, Research Diets, Inc., New Brunswick, NJ, USA) fortified with either vitamin A as retinyl acetate (Sigma-Aldrich, St. Louis, MO, USA) or *Dunaliella bardawil* powder (Nikken Sohonsha, Gifu, Japan) containing ~10% of the dry weight as BC [13]. Detailed diet formulation is listed in Appendix A. To prepare the feed, 250 mL warm distilled water was mixed with 12 g fish gelatin until the solution became clear. Then, the gelatin solution was thoroughly mixed with 1 kg of feed and 4.5 mg of vitamin A (dissolved in 40 μL olive oil) or 80 g *Dunaliella bardawil* powder. The feed mixtures were poured into containers and stored at −20 °C. Since BC isomers are oxidized upon exposure to air and light, the BC content and ATBC/9-cis ratio in *Dunaliella bardawil* powder were examined before adding the alga to the feed. The feed was replaced every 2 days.

### 4.3. Study Design

The mice were allocated into two groups, 15 animals per group, while ensuring that the initial body weight variation was similar in each group. In each diet group, the mice were housed in two separate cages (6–8 animals per cage). The mice were fed for 23 weeks with either one of two high-fat diets: I. HFD, containing 4.5 mg vitamin A per kg feed, or II. HFD-BC containing 6–8 g BC (~50% all-trans and ~50% 9-cis) per kg feed. At the most, body weight and feed intake were recorded every 10 and 4 days, respectively. Body composition was determined in anaesthetized mice at the 14th week using a TD-NMRLF50 minispec Live Mice Analyzer (Bruker Optics, Billerica, MA, USA). The NMR instrument was calibrated according to the manufacturer’s instructions, and the mice were weighed and inserted into the test chamber.

### 4.4. Carotenoid and Retinol Analysis

All-trans retinol (vitamin A) and BC isomer levels in plasma, liver (*n* = 5), and adipose tissue (*n* = 5) were determined by HPLC, as previously described by Harari et al. [29].

### 4.5. Analysis of Gene Expression by Real-Time PCR

An RNeasy Lipid Tissue Mini Kit (QIAGEN, Hilden, Germany) was used to extract RNA from eWAT (*n* = 5–6) and iBAT (*n* = 5–7). The extracted RNA quantity and quality were determined using NanoDrop One (Thermo Scientific, Wilmington, DE, USA), and the RNA was stored at −80 °C. Equivalent amounts of the total RNA were reversely transcribed to cDNA using the High-Capacity cDNA Reverse Transcription Kit (Applied Biosystems, Waltham, MA, USA). Quantitative real-time PCR was performed in duplicates with the 7500 Real-Time PCR (Applied Biosystems, Waltham, MA, USA), FastStart Universal Probe Master ROX (Roche, Pleasanton, CA, USA), probes labeled at the 5′ end with fluorescein (FAM) and at the 3′ end with a dark quencher dye (Universal ProbeLibrary, Roche, Indianapolis, IN, USA) and custom primers (Sigma-Aldrich, St. Louis, MO, USA). A predesigned qPCR assay (PrimeTime^®^ Mini 135qPCR Assay, Integrated DNA Technologies, Coralville, IA, USA) was used for the quantification of *Pparγ* expression. All probes and primers are listed in the Appendix A). Relative quantification was done using the 2^−ΔΔCT^ method against control glyceraldehyde-3-phosphate dehydrogenase (*Gapdh*) [42]. The results are expressed as fold-change relative to the HFD group.

### 4.6. Adipose Tissue and Liver Histological Analysis

eWAT and liver samples were fixed in 4% formaldehyde buffered solution for 72 h and embedded in paraffin. Tissue sections (5 μm) were stained with hematoxylin and eosin (H&E) and visualized under a microscope at 20× (Olym-144pus BX51 microscope, Olympus UPLanApo 20×/0.70 objective, Nikon DS-Fi1 camera, Olympus, Tokyo, Japan). Initial processing of the eWAT image sections was performed using AdipoCount software [43]. Then, the sizes and numbers of the adipocytes were measured using Fiji software [44] with Adiposoft plugin [45] in manual editing mode. Blinded scoring of the liver sections was performed by the Head of the Institute of Pathology at Sheba Medical Center (Prof. Iris Barshack), according to NAFLD activity score criteria [46,47]. Macrophage infiltration in adipose tissue was estimated by CD68 staining [48].

### 4.7. Analysis of Plasma Parameters

During the 18th week, retro-orbital blood samples were collected after a 4 h fast into collection tubes containing EDTA for insulin measurement, whereas terminal blood samples (23rd week) were used to test all other parameters. We used a colorimetric enzymatic procedure to measure the total plasma cholesterol and triglycerides (AU480 chemistry analyzer, Beckman Coulter, Inc, Brea, CA, USA). Plasma insulin (MRC-10-1249-01, Mercodia, Uppsala, Sweden), leptin (MOB00, R&D Systems, Inc., Minneapolis, MN, USA), and adiponectin (MRP300, R&D Systems, Inc., Minneapolis, MN, USA) levels were measured using commercial ELISA following the manufacturer’s protocols.

### 4.8. Intraperitoneal Glucose Tolerance Test and Fasting Glucose

An intraperitoneal glucose tolerance test (IPGTT) was performed during the 16th week of treatment by injecting weight-adjusted volumes of 20% (*w*/*v*) glucose solution (2 g glucose/kg body weight) after 4 h of fasting. IPGTT blood samples were measured at 0, 15, 30, 60, and 120 min. Fasting blood glucose (4 h) was assessed during the 18th week. All blood samples were collected from the tail vein, and glucose levels were measured using a glucometer (FreeStyle Lite, Abbott, Alameda, CA, USA).

### 4.9. Statistical Analyses

Data are expressed as the means ± SEM. Statistical significance was defined as *p* < 0.05. Differences in body weight between the HFD and HFD-BC groups were analyzed by a mixed effects analysis, followed by Bonferroni’s multiple comparisons test. Other differences between the two groups were analyzed by an unpaired Student’s *t*-test or Mann–Whitney test (NAFLD activity score data). BC concentrations in tissues (WATs, BAT, liver) of the HFD-BC group were tested by 1-factor ANOVA followed by Tukey’s multiple comparisons test, or by a Brown–Forsythe ANOVA followed by Tamhane’s T2 multiple comparisons test. Statistical analysis and figures were generated using GraphPad Prism 8.

## Figures and Tables

**Figure 1 marinedrugs-20-00433-f001:**
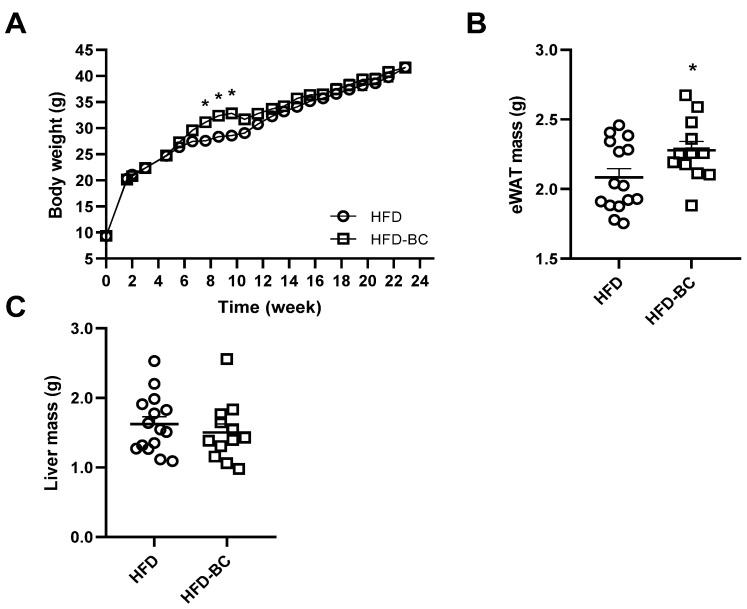
Three-week-old male mice were fed an HFD (*n* = 15) or an HFD-BC (*n* = 13) for 23 weeks. (**A**) Body weight throughout the experiment. (**B**) eWAT mass. (**C**) Liver mass. Values are means ± SEM. * Different from HFD group, *p* < 0.05. BW, body weight; eWAT, epididymal white adipose tissue; HFD, high-fat diet; HFD-BC, high-fat diet supplemented with *Dunaliella bardawil*.

**Figure 2 marinedrugs-20-00433-f002:**
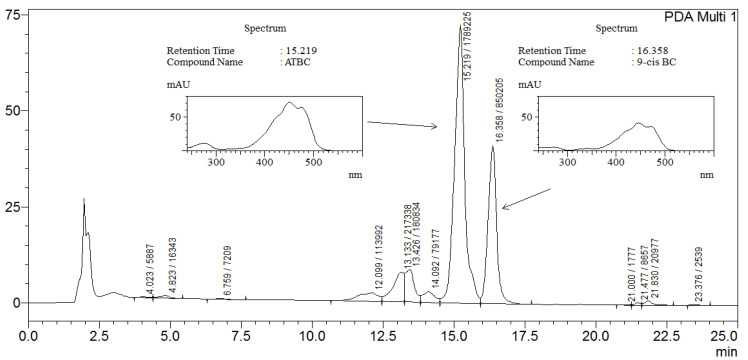
Representative HPLC analysis of hepatic carotenoids (spectrum range 200–700 AU) in a liver of a mouse in the HFD-BC group.

**Figure 3 marinedrugs-20-00433-f003:**
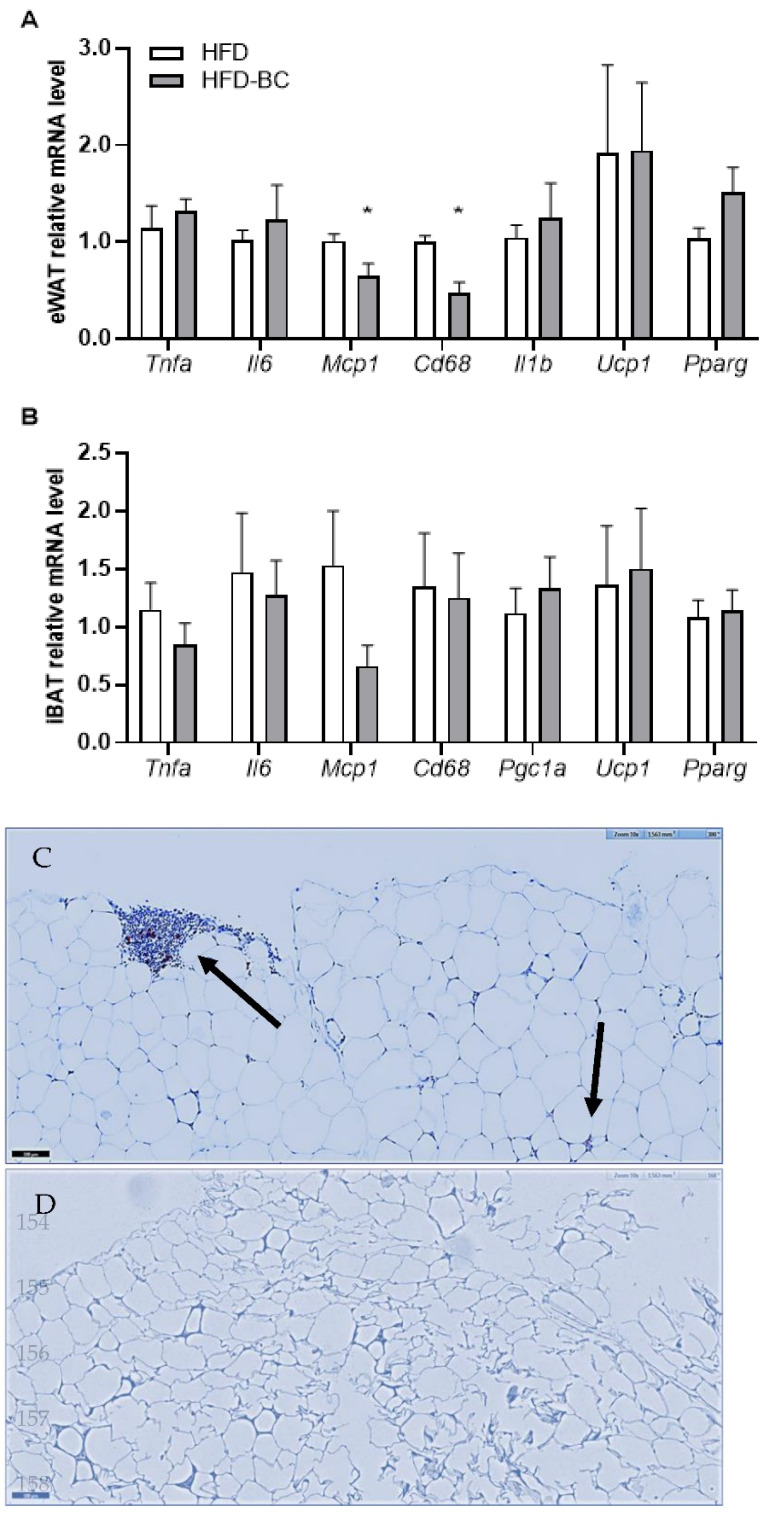
eWAT (**A**) and iBAT (**B**) relative mRNA levels (*n* = 5–7). Gapdh was used as a reference gene after 23 weeks. Histochemical detection of macrophages in epididymal adipose tissue of HFD (**C**) and HFD-BC (**D**); the bar = 100 micrometer. Black arrows indicate macrophages. Values are mean ± SEM. * Different from HFD group, *p* < 0.05.

**Figure 4 marinedrugs-20-00433-f004:**
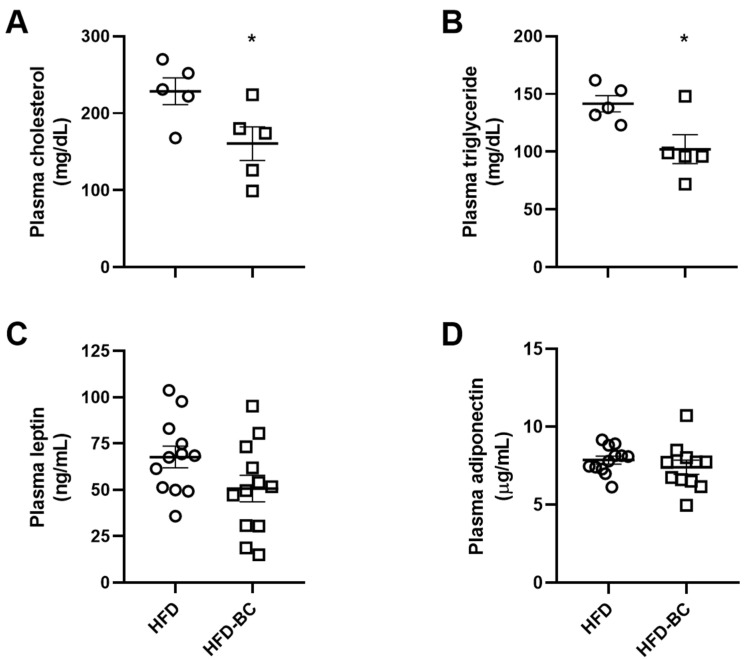
Fasting cholesterol (**A**), triglycerides (**B**), leptin (**C**), and adiponectin (**D**) concentrations in the plasma after 23 wk of treatment (A–B *n* = 5, C–D *n* = 11–12). Values are mean ± SEM. * Different from HFD group, *p* < 0.05.

**Table 1 marinedrugs-20-00433-t001:** Tissue and plasma retinol levels in mice fed an HFD or an HFD-BC for 23 weeks ^1^.

	HFD (*n* = 5)	HFD-BC (*n* = 5)
Liver retinol, μg/g tissue	126 ± 15.8	194 ± 15.8 *
eWAT retinol, μg/g tissue	0.21 ± 0.04	0.37 ± 0.03 *
iWAT retinol, μg/g tissue	0.21 ± 0.01	0.50 ± 0.07 *
iBAT retinol, μg/g tissue	0.25 ± 0.04	0.41 ± 0.03 *
Plasma retinol, μg/mL	0.19 ± 0.01	0.16 ± 0.03

^1^ Data are presented as mean ± SEM. * Different from HFD group, *p* < 0.05. iBAT, interscapular brown adipose tissue; iWAT, inguinal white adipose tissue.

**Table 2 marinedrugs-20-00433-t002:** Adipose tissue β-carotene levels, isomer ratios, and β-carotene/retinol ratios in mice fed an HFD-BC for 23 weeks ^1^.

	eWAT (*n* = 5)	iWAT (*n* = 5)	iBAT (*n* = 5)
Total BC, μg/g tissue ^2^	0.34 ± 0.05b	0.41 ± 0.04b	2.13 ± 0.17a
ATBC/9-cis BC ratio	6.67 ± 1.19b	4.91 ± 0.54b	10.6 ± 0.61a
BC/retinol ratio ^2^	0.95 ± 0.15b	0.86 ± 0.10b	5.20 ± 0.46a

^1^ Adipose tissue BC levels were measured by HPLC analysis. ^2^ Total BC is the sum of ATBC and 9-cis BC concentrations. Data are presented as means ± SEM. Labeled means in a row without a common letter differ, *p* < 0.05. ATBC, all-trans β-carotene; BC, β-carotene.

## Data Availability

Not applicable.

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
