# Peer review of "β-Carotene from the Alga Dunaliella bardawil Decreases Gene Expression of Adipose Tissue Macrophage Recruitment Markers and Plasma Lipid Concentrations in Mice Fed a High-Fat Diet"

_marinedrugs, 2022, doi:10.3390/md20070433_

Round 1

Reviewer 1 Report

Manuscript entitled: β-carotene from the alga Dunaliella bardawil decreases gene expression of adipose tissue macrophage recruitment markers and plasma lipid concentrations in mice fed a high-fat diet

The manuscript incorporates new elements on the effect of the incorporation of beta-carotene in the diet, however, some doubts that arise from the results should be explained in greater detail, and are mentioned below:

  1. Include a brief explanation of the models fatty liver in Ldlr-/-, Apoe-/-, and db/db mice models.
  2. What can be the explanation of the following results:

Expectedly, the accumulation of total BC (μg/g tissue) in the liver

(7.04±1.45) of the HFD-BC group was 20-fold, 17-fold, and 3-fold greater than eWAT,

iWAT, and iBAT, respectively (P<0.001). ATBC/9-cis BC ratio in the liver (1.82±0.19) was

lower by 3.7-fold, 2.7-fold, and 5.8-fold than in eWAT, iWAT, and iBAT, respectively, here

suggesting that the liver stores more 9-cis BC than the adipose tissue

What are the possible mechanisms for this?

  1. Explain deeply Table 2
  2. Why are Mcp1 and Cd68 only significantly different in eWAT and not in IBAT?

  1. In relation to the results of leptin and adiponectin:

Even though they are different functions, different signaling pathways, and opposite effects on inflammation, it is important to explain: Why it reduces leptin but has no effect on adiponectin?

  1. What can be the explanation of the following results:

We have demonstrated that dietary BC maintains tissue and plasma vitamin A reservoir and that BC accumulation in BAT was elevated compared with WAT. Unexpectedly, we found that BC supplementation increased eWAT mass but reduced eWAT

mRNA levels of macrophage recruitment markers and lowered plasma cholesterol and

triglyceride levels in high-fat diet-fed mice. Contrary to our hypothesis, BC treatment did

not affect the mRNA levels of the genes involved in regulating adipose tissue thermogenic

activity in iBAT or eWAT.

  1. If in the mouse, lack the BC degradation enzyme β-Carotene oxygenase 1 (Bco1),

How is that dietary BC maintains tissue and plasma vitamin A reservoir?

Considering that: To the best of our knowledge, the estimated efficiency of BC isomer conversion to vitamin A in mice has not been evaluated.

Reviewer 2 Report

The manuscript entitled “β-carotene from the alga Dunaliella bardawil decreases gene expression of adipose tissue macrophage recruitment markers and plasma lipid concentrations in mice fed a high-fat diet” described the effect of BC-rich Dunaliella bardawil powder as the sole source of vitamin A on the development of obesity and its complications in mice. On the basis of this study, I think the manuscript is not acceptable in its present form and requires major revision.

Major points

  1. The authors did not describe the study design clearly, especially diet constitutes. Why HFD contain 4.5 mg Vitamin A but HFD-BC group contain 6-8 g BC? The author should explain it more. Additionally, the dose of BC in diet should not be a range (6-8 g), it must be precise.
  2. Figure 1 showed that the HFD group and HFD-BC group had 15 and 13 mice, respectively. But in other figure or table, the mice in each group was reduced from 15 to 5-7. Why?
  3. The present study found that dietary BC increased eWAT, which was inconsistent with other studies. The authors should discuss it more detail.
  4. In Table 1, why dietary BC did not increase plasma retinol level?
  5. In Table 2, the higher levels of BC accumulated in iBAT that that of WAT was observed. Why?
  6. The authors declared that dietary BC might reduce adipose tissue macrophage recruitment according to the results of MCP-1 and CD68 mRNA levels. I think more experiments such as IHC staining should be done to further confirm the phenomenon. What’s more, why the mRNA levels of pro-inflammatory cytokines such TNF-α, IL-1β and IL-6 were not decreased in HFD-BC group, since diet BC reduced adipose tissue macrophage recruitment.

Minor points

  1. Line 70-72. The sentence was incomplete.
  2. Line 81, Figure1B-D should be changed into Figure1C.
  3. In Table 2, Total BC, μg/g tissue2 and BC/retinol ratio2. What does 2 mean?

Round 2

Reviewer 1 Report

According to the answers and modifications made to the manuscript.

Reviewer 2 Report

The authors addressed all questions and concerns. There are no additional problems.